# Why Do New Parents Stop Practising Sport? A Retrospective Study towards the Determinants of Dropping Out after Becoming a Parent

**DOI:** 10.3390/ijerph21030342

**Published:** 2024-03-14

**Authors:** Hidde Bekhuis, Jasper van Houten, Femke van Abswoude

**Affiliations:** 1Department of Orthopedagogics: Learning, Education and Development, Behavioural Science Institute, Radboud University, 9104 HE Nijmegen, The Netherlands; femke.vanabswoude@ru.nl; 2School of Sport & Exercise, HAN University of Applied Sciences, 6525 AJ Nijmegen, The Netherlands; jasper.vanhouten@han.nl

**Keywords:** parenthood, sport participation, drop out, socialisation, social support

## Abstract

Despite the known health benefits of sport, recent studies showed that parenthood is related to decreased sport participation. Changes in sport behaviour after becoming a parent have been explained by gender or with the rational resource perspective of limited time and energy. However, the latter is mostly theoretical, since empirical insights on resource mechanisms are scarce. We want to improve and go beyond these explanations by investigating them empirically and by examining sport socialisation during the formative years as an alternative explanation. Consequently, our main objective is to explain changes in sport participation after becoming a parent with gender, limited resources and socialisation with sport. To this end, we employ representative Dutch survey data of new parents (*n* = 594), containing detailed information on sport careers and sport socialisation, as well as babysitter availability, partner support and physical discomfort after childbirth. The results of the logistic regression analyses show that, besides gender and resource mechanisms, sport socialisation and social support seem to have a great impact on sport behaviour when people become parents. That is, men are more likely to continue sport participation, as well as people with more resources (physical, temporal and social) and more socialisation with sport during the formative years. So including sport socialisation and social support seems necessary to better explain and prevent sport dropout during major life transitions, like becoming a parent.

## 1. Introduction

### 1.1. Background

Participating in physical activity and sport has been shown to have positive effects on physical health, as well as social and mental well-being [1]. Despite these benefits, our society has reached a pinnacle in physically inactive behaviour, affecting all age groups and people from all socioeconomic levels [2]. This is why the WHO has declared the current trend towards an increasingly inactive world population as a pandemic [3]. In spite of numerous efforts from scientists, professionals, sport organizations and policymakers, the pandemic of inactivity is still ongoing and even worsening [4]. This highlights that the sustainability of active behaviour should be the main target to advance more active lifestyles. A way to study this is not to focus on why people start practising sport but instead to focus on why people drop out.

An important aspect of an active lifestyle in Western societies is sport participation [5,6,7,8]. Empirical evidence suggests that life transitions can have a negative impact on sport participation, particularly the transition to adulthood [9,10,11,12]. During this transition—marked by major life events such as going from primary to secondary school, starting a study, getting married, and becoming a parent—the likelihood of stopping practising sport increases, posing a threat to sustainable sport participation during the life course [11,13,14].

Theoretically, this can be explained by framing it in the neo-Weberian tradition of looking at social action from a resource perspective [15,16,17,18]. Based on this perspective, the opportunities and hindrances in accessing valued outcomes, in this case sport participation, are determined by individual resources [19]. Major life events bring forth new roles and responsibilities, leading to changes in temporal, social, physical, mental and economic resources [12,20,21,22]. These changes in resources, such as alterations in leisure time, social capital and physical or mental well-being, may trigger periods of socioeconomic adaptation and readjustment, potentially leading to behavioural changes, including shifts in sport participation patterns [11,13,23]. Thus, changes in sport behaviour throughout the life course, including dropping out during the transition to parenthood, can be interpreted as adaptations to new configurations of limited resources for sport participation after experiencing these major life events [13,24,25,26].

However, this theoretical explanation of the impact of major life events on (stopping) sport participation is still largely unexplored [13,24,25,26]. The studies that have empirically tested this explanation highlight its value in understanding patterns in sport participation but also leave a significant part of the variation unexplained. Consequently, the objective of this study is to gain more empirical insight into both the resource mechanisms as well as possible alternative or additional explanations for changes in sport participation following major life transitions. This is indispensable for understanding why people stop practising sport when confronted with major life events [11] and essential for the development of interventions to prevent dropping out, in particular during the transition to adulthood.

### 1.2. Research Question

We want to improve and go beyond the theoretical explanation from a resource perspective by focussing on dropping out of sport after becoming a parent and investigate empirically how this is affected by different resources for sport participation: physical, temporal and social resources. In addition, we will include the influence of sport socialisation during the formative years as an alternative explanation [27]. Consequently, our question reads: Can stopping sport participation after becoming a parent be explained by the presence or absence of resources for sport participation and socialisation with sport? 

We focus on becoming a parent, because this is a major life event that marks the transition to adulthood, and empirical evidence shows it has a significant impact on (stopping) sport participation [11,13,28,29]. Additionally, becoming a parent is a relevant life event to empirically test the resource mechanisms, as it can alter the temporal, physical and social resources needed to practise sport that might explain dropout, like leisure time, energy, recovery from giving birth, partner support and availability of a babysitter. Lastly, we include sport socialisation during the formative years as well, because earlier research shows this can affect sport participation during the life course [27,30,31,32]. By combining the resource perspective with socialisation, we are one of the first that will test both mechanisms simultaneously. To answer the research question, we employ representative Dutch survey data of 594 new parents containing detailed historical information on their sport careers and sport socialisation, as well as leisure time and babysitter availability, partner support and physical conditions after childbirth.

### 1.3. Theory and Hypotheses

#### 1.3.1. Physical Resources: Being Physically Able to Practise Sport after Becoming a Parent

When sport participation is viewed from a resource perspective [15,16,18], becoming a parent can reduce the resources available for sport participation. First, parenthood can lead to a decrease in the physical condition required for sport [33,34], which decreases the possibility to participate in sport. Especially for women, the transition to parenthood has a considerable physical impact. That is, the process of pregnancy and childbirth that women go through is physically very demanding. Therefore, we expect that, in general, women are more likely than men to stop doing sport after becoming a parent (Hypothesis 1).

Although from a gynaecological point of view women can be recovered after six weeks after giving birth [35], it could take much longer to be fit enough to participate in sport again [36]. This leads us to expect that the likelihood of stopping practising sport is specifically higher for woman who need more time to physically recover from giving birth (Hypothesis 2).

Besides pregnancy and giving birth, parenthood as a whole can be physically demanding and stressful for both mothers and fathers [34]. New parents often experience lack of and disturbed sleep and have to do laborious caring tasks (e.g., carrying, cradling or rocking the baby). However, the extent to which this results in less physical resources to practise sport depends on several factors, such as how ‘easy’ or ‘difficult’ a baby is, how long the period of poor sleep lasts and how this is experienced by the parents. Consequently, we expect that people who (still) feel fatigued while being a parent are more likely to stop practising sport (Hypothesis 3).

#### 1.3.2. Temporal Resources: (Not) Having Enough Time to Practise Sport after Becoming a Parent

Another important resource for sport participation is having time to practise sport. After the birth of a child, the new parents are faced with new childcare responsibilities. This introduces caring tasks that take time, which can cause time pressure [34,37]. The experience of time pressure results in people ceasing informal activities when more formal and obligatory tasks require attention, such as childcare [13,38]. Practising sport is such an informal and optional activity, which can be given up without too many direct consequences. This leads us to expect that the likelihood of stopping practising sport after becoming a parent is higher for people who do not have enough time to practice sport (Hypothesis 4).

#### 1.3.3. Social Resources: Having Social Support to Practise Sport after Becoming a Parent

The extent to which the transition into parenthood makes new parents have less time for sport can be affected by their social network [13]. The partner as well as babysitter availability seem to affect sport participation [19,39,40].

Having a partner can make (child) care tasks lighter. After all, care tasks can be divided between two people [41,42]. Sharing care responsibilities can provide the opportunity to do things for oneself and spend time on optional or leisure activities, such as sport. However, childcare tasks are often not equally distributed between partners [43,44]. A partner therefore only enables one to practise sport when the partner actually takes over care tasks and allows the other to do things for him or herself [13]. Consequently, we expect that the likelihood of stopping practising sport after becoming a parent is higher for people who do not have a supporting partner (Hypothesis 5).

Another social resource that enables new parents to continue to practise sport is a babysitter. By hiring a babysitter, parents buy time they do not have to spent on childcare tasks. This time can be spent on sport [40]. This leads us to expect that the likelihood of stopping practising sport after becoming a parent is higher for people who do not have a babysitter (Hypothesis 6).

#### 1.3.4. Socialisation with Sport

The (lack of) resources for practising sport associated with becoming a parent cannot explain completely whether new parents continue their participation in sport [11,13,26]. Another mechanism that can solve the puzzle on why people continue or stop doing sport when they become a parent is socialisation [45]. Socialisation with sport includes the habits, norms and expectations that an individual holds concerning sport and is to a large part developed during the formative years. Socialisation with sport, including the habit of sport participation during childhood and adolescence, is seen as an important factor why people continue doing sport when they become older [31,32]. Although socialisation is far more complex than just ‘teaching’ people how to behave [30], it has been suggested that practising sport during the formative years is an important contributor for sport participation at a later age [46]. Although there is, to our knowledge, no research that has examined the relationship between sport participation during childhood and the continued participation as a parent, we assume that socialisation with sport influences the sport behaviour of people who make the transition into parenthood. Consequently, we expect that the likelihood of stopping practising sport after becoming a parent is lower for people who practised sport in their last two years at primary school or in their last two years at secondary school (Hypothesis 7a/b).

Socialisation can take place not only through the experiences and habits of the individual but also through the context in which they find themselves. One of the most important contexts for children is the family in which they grow up [47]. As a person grows older, the influence of the family, and therefore the father and mother, decreases, and actors outside the family become more important [30]. Due to the time spent with the family and the importance of the parents during (early) childhood, the extent to which parents participate in sport affects people’s sport participation in later life positively [31], by being a role model either as a coach or by participating in sport themselves. We hypothesize that the likelihood of stopping practising sport after becoming a parent is lower for people who had sporty parents in their last two years at primary school or in their last two years at secondary school (Hypothesis 8a/b)

#### 1.3.5. Overview of the Hypotheses

**Hypothesis** **1.**
*Women are more likely than men to stop doing sport after becoming a parent.*


**Hypothesis** **2.**
*The likelihood of stopping practising sport is specifically higher for women who need more time to physically recover from giving birth.*


**Hypothesis** **3.**
*People who (still) feel fatigued while being a parent are more likely to stop practising sport.*


**Hypothesis** **4.**
*The likelihood of stopping practising sport after becoming a parent is higher for people who do not have enough time to practice sport.*


**Hypothesis** **5.**
*The likelihood of stopping practising sport after becoming a parent is higher for people who do not have a supporting partner.*


**Hypothesis** **6.**
*The likelihood of stopping practising sport after becoming a parent is higher for people who do not have a babysitter.*


**Hypothesis** **7.**
*The likelihood of stopping practising sport after becoming a parent is lower for people who practised sport (a) in their last two years at primary school or (b) in their last two years at secondary school.*


**Hypothesis** **8.**
*The likelihood of stopping practising sport after becoming a parent is lower for people who had sporty parents (a) in their last two years at primary school or (b) in their last two years at secondary school.*


## 2. Materials and Methods

### 2.1. Data and Respondents

The present study used the TRansitions Into Active Living (TRAIL) wave 1 dataset [48]. This dataset is based on a sample of the I & O Research panel. This panel consists of more than 37,000 active respondents and is based on probability samples of all Dutch municipalities, making the panel representative of the Dutch population. Respondents receive financial compensation for cooperation based on the number of questions they have to answer. For this data collection, 10,140 respondents age 16 to 40 years were invited. Finally, 4691 (46.26%) respondents filled out the questionnaire. People who finished the questionnaire in less than 5 min or who showed patterns of straightlining were excluded from the analyses (*n* = 76; 1.62%). Moreover, since we are only interested in people who had already become a parent, respondents who did not have children were excluded as well (*n* = 3522; 76.32%), just as people who did not practise sport before they became a parent (*n* = 404; 37.03%). Furthermore, since we asked respondents questions about their sport behaviour in the year that their first child was 1 year old, we excluded respondents who had a first child of less than 1 year old (*n* = 61; 8.89%). We did so to exclude the newborn phase, which is generally very hectic, and because respondents were questioned about their sport behaviour and resources for sport participation in the year in which their firstborn child was 1 year old. Finally, respondents who did not answer all questions measuring the independent variables were excluded as well (*n* = 33; 5.09%). Consequently, the analyses were conducted on 594 respondents who were parents and who participated in sport before they made the transition into parenthood.

### 2.2. Measurements

Table 1 shows the descriptive statistics of all variables. Below we discuss the construction of these variables.

#### 2.2.1. Dependent Variable: Stop Practising Sport

To measure whether people stop practising sport after becoming a parent, we made use of two questions. First, we asked ‘Did you play sport in the year before having your first child’? This question could be answered with no or yes. Next, we asked ‘Did you do sport in the year your first child was 1 year old’?, with the same response categories. Due to our initial selection criteria, all respondents (*n* = 594) in our data indicated they practised sport before they became parents. Three hundred ninety-one respondents (65.82%) indicated they practised sport in the year their first child was 1 year old, and 203 respondents (34.18%) indicated that they stopped practising sport after they became parents.

#### 2.2.2. Independent Variables

To measure physical and temporal resources for sport participation, respondents were presented with multiple reasons for practising sport more or less in the year that their first child was 1 year old compared to what they did before they became a parent and were asked which reason(s) applied to them. One of these reasons was ‘I started exercising less because I wasn’t physically able’, which we used to measure whether or not women were physically recovered after giving birth to be able to participate in sport. All men automatically scored 0 on this variable. Seventy-two women (12.12%) of the respondents who indicated this reason applied to them scored 1, and all other women scored 0. To measure whether or not respondents were too tired to practise sport in the year the first child was 1 year old, we used the reason ’I started exercising less because I was too tired’: 28.62% of the respondents indicated that this reason applied to them. Not having enough time was measured with the reason ‘I started exercising less because I didn’t have time to practice sport’, which applied to 52.53% of the respondents.

To measure social resources for practising sport, we firstly determined whether or not respondents had a supporting partner by asking ‘In the year that your first child was 1 year old, did the division of care responsibilities with your partner allow you to do things for yourself, such as shopping, practising sport and going to the cinema’? Respondents could answer in four categories: ‘no, I didn’t have a partner’, ‘no, my partner didn’t enable me to do the things I wanted to do’, ‘yes, my partner enabled me to do some things but not as much as I wanted’ and ‘yes, my partner enabled me to do almost everything I wanted’. We made a dichotomous distinction between (0) no or not enough support by the partner (the first three categories, 57.24%) versus (1) enough support (yes, my partner enabled me do almost everything I wanted, 42.76%). Secondly, we determined if respondents had a babysitter or nanny by asking the question ‘In the year that your first child was 1 year old, did you have a babysitter outside your (possible) working hours? Respondents could indicate if they had no babysitter at all, that they had a babysitter who came occasionally, that they had a babysitter who came at set times or that they had a (near) full-time nanny or au pair. Because of the distribution of the response categories, we make the dichotomous distinction between (0) having no babysitter or only occasionally (74.58%) and (1) having a babysitter at set times or an au pair (25.42%). (We performed robustness checks with other compositions. This did not lead to different results.)

Furthermore, as an indication for socialisation with sport we first determined whether or not respondents practised sport at the primary school and secondary school level by asking the following two questions: ‘Did you practise sport in grade 7 and 8 of primary school’? (Grade 7 and grade 8 are the last two years of primary school in the Netherlands) and ‘Did you practise sport in the last two years of secondary school’? In grade 7 and 8 of primary school, 76.77% of the respondents practised sport, while 64.98% of the respondents did so in the last two years of secondary school. Second, we measured whether or not respondents had parents who participated in sport at primary and secondary school by asking ‘Did both your parents, or one of your parents, practise sport at a sport club when you were in grade 7 and 8 of primary school’? and ‘Did both of your parents, or one of your parents, practise sport at a sport club when you were in the last two years of high school’? In the last two years of primary school and secondary school, 36.87% and 33.50% of the respondents, respectively, had parents who practised sport at a sport club.

#### 2.2.3. Personal Characteristics

Finally, we controlled for personal background characteristics that are important for explaining sport participation [19,49]. We included the age of the respondents when they became a parent for the first time, ranging from 18 to 38 with a mean of 28.87 and SD of 3.35. Furthermore, we indicated if a respondent is a male (31.99%) or female (68.01%). In addition, we included if respondents have a Dutch (90.07%) or a non-Dutch (9.93%) ethnicity. Finally, educational level was determined as the highest completed educational level in the year the questionnaire was administered. We included this educational level as the theoretical age respondents finished their education, ranging from 12 (primary education) to 23 (master’s degree), with a mean of 20.90 and SD of 1.99.

### 2.3. Data Analysis

Our dependent variable ‘stop practising sport’ is a dichotomic (0 1) variable; hence to test our hypotheses we used the multiple logistic regression command ‘logit’ in STATA 17.0. All the statistical tests were regarded as statistically significant, with a *p*-value less than 0.05 (two-sided).

We present odds ratios (Exp(B)). The odds ratio (OR) is the multiplication factor of the odds and signifies the increase or decrease in the odds per unit of an independent variable. Essentially, an OR above (below) 1 indicates that the odds of stopping doing sport increases (decreases) if the value of an independent variable increases (decreases).

In Model 1 we included the personal characteristics, the physical resources ‘still recovering from giving birth’ and ‘too tired’ and the temporal resource ‘not enough time’. In the second model we added the social resources ‘no babysitter’ and ‘a supportive partner’. In the last model we added the socialisation with sport variables ‘sport during primary school’, ‘sport during secondary school’, ‘sporty parents during primary school’ and ‘sporty parents during secondary school’.

## 3. Results

Table 2 shows the results of the logit regression analyses. Although we included the variables in three steps, the effects of already included variables did not change significantly by adding new variables, meaning none of the added variables (completely) explains the previously included variables. Therefore, we only discuss the full model, Model 3.

### 3.1. Personal Characteristics

As predicted by Hypothesis 1, women are (almost) twice as likely to stop practising sport after becoming a parent than men (OR = 0.532; *p* = 0.003). In addition, the age at which people become a parent does not have a significant effect on the likelihood of stopping practising a sport after becoming a parent. Moreover, people with a non-Dutch nationality are more than two times more likely to stop practising sport after becoming a parent than those with a Dutch nationality (OR = 0.421; *p* = 0.003). Finally, every extra year of education decreases the odds of stopping practising sport after becoming a parent by 11.6% (OR = 0.884; *p* = 0.008).

### 3.2. Physical Resources

Mothers who were not physically recovered from giving birth are three times (OR = 3.117; *p* < 0.000) more likely to stop participating in sport. This finding is in accordance with Hypothesis 2. (We checked if the gender effect remains if we only include women who are recovered from giving birth. This does not change the gender effect that women are more likely to stop than men).

In addition, parents who feel too tired are 71.9% more likely to stop participating in sport than parents who do not feel too tired (OR = 1.719; *p* = 0.006), which is in accordance with Hypothesis 3.

### 3.3. Temporal Resources

In accordance with Hypothesis 4, it appears that not having enough time increases the likelihood of stopping practising sport after becoming a parent by 162.3% (OR = 2.623; *p* < 0.000).

### 3.4. Social Resources

As predicted by Hypothesis 5, having a supportive partner decreases the odds of stopping practising sport after becoming a parent. The odds of stopping practising sport were 37.2% smaller (OR = 0.628; *p* = 0.018) for new parents who had a supporting partner compared to new parents who did not have a partner or whose partner was not supportive.

Hypothesis 6 expected that new parents who did not have a babysitter are more likely to stop practising sport. However, we found no significant differences in the likelihood of stopping practising sport between new parents who did or did not have a babysitter in the year their first child was 1 year old.

### 3.5. Socialisation in Sport

We predicted that the likelihood of stopping practising sport after becoming a parent is lower for people who practised sport in their last two years of primary school (Hypothesis 7a) or in their last two years of secondary school (Hypothesis 7b). We only found evidence for socialisation with sport during the last two years of secondary school. The odds of stopping practising sport were 42.9% smaller for new parents who practised sport during the last two years of secondary school compared to those who did not practise sport then (OR = 0.571; *p* = 0.005).

In Hypothesis 8a,b, we predicted that the likelihood of stopping practising sport after becoming a parent is lower for those who saw their parents participate in sport during the last two years of primary school (8a) and the last two years of secondary school (8b). We only found support for the first hypothesis. The odds of stopping practising sport were 56.4% smaller for new parents who had parents who participated in sport when they were in primary school, compared to those who did not have sporty parents then (OR = 0.436; *p* = 0.006).

## 4. Discussion

This paper adds to the existing literature by focussing on dropping out of sport after becoming a parent and investigating empirically how this is affected by different resources for sport participation, including physical, temporal and social resources, as well as socialisation with sport. Corroborating earlier studies [11,13,26,30,31,32,33,34,35,36,37,38], we showed that both resources and socialisation affect the likelihood of new parents stopping practising sport in the year their child is 1 year old. As an extension of these earlier studies, we showed that, taking all these factors together, resources and socialisation simultaneously affect sport participation of new parents. Physical resources appear to be the most important, while the other resources and socialisation affect sport participation to more or less the same extent. (This can be concluded from the fact that the OR are in the same order of magnitude.) Below we will discuss these findings in more detail.

The resource perspective is one way of explaining the decrease in sport participation after becoming a parent [15,16,17,18]. The results of this study provide clear evidence that an experienced decrease in resources is a reason for new parents to stop participating in sport. First, based on the physical resources, we showed that physical recovery from childbirth decreases sport participation for women. In addition, sport participation of women is more affected compared to men. These findings are in accordance with previous studies [25,46,50] and could reflect the fact that not only the physical impact of becoming a parent is greater for women than for men but also that prevailing traditional gender roles and norms associated with parenthood influence sport participation [43,44], as well as the gender norms regarding sport participation [51]. In addition, we found that parents who are too tired (or perceive to be so) drop out of sport more often. Together, these results highlight the importance of physical resources for sport participation after becoming a parent.

In addition to the physical resources, the results also provide evidence that temporal resources are important. In accordance with previous studies, it appeared that new parents who perceive to have less time for sport are more likely to stop [13,52]. This finding corroborates the argument that new parents experience that more formal and obligatory tasks require the limited time available, leading to the cessation of informal activities like sport.

Thirdly, we investigated social resources. Here the results were mixed. We argued that social resources could support a new parent in continuing their sport participation by taking care of the child(ren). However, we only found that having a supportive partner reduced the chances of stopping participating in sport, whereas having a babysitter did not affect this chance at all. Despite these mixed results, it is still plausible that social resources are important for continuing sport participation for people who become parents. However, the role of a babysitter may be different than what we expected. Narratives regarding sport participation after becoming a parent in the qualitative study of Van Houten [13] showed that practising sport was not an activity for which many people were willing to arrange a babysitter. It may be more probable that parents who participate in sport plan this together with their partner, which is in line with the first results, or when their child is going to daycare. So this might be why having a babysitter (or not) does not affect stopping after becoming a parent. In addition, other social resources that were not included in this study, like friends that encourage one to sport or that sport with someone or other family members that enable sport participation by taking care of the children, may be factors that should be taken into account in future work to better understand the role of social resources.

Another mechanism that may explain why new parents do or do not continue sport participation is socialisation. We found clear evidence that being socialised with sport participation decreases the likelihood to stop doing so when someone becomes a parent for the first time. However, our results show that this effect is more complicated than just participating in sport at all ages and seeing one’s parent(s) sport at all ages. First of all, whereas practising sport during the last two years of secondary school clearly reduces the likelihood of stopping practising sport after becoming a parent for the first time, practising sport during the last two years of primary school does not have an effect. A possible explanation why practising sport at primary school has no influence is lack of variation: 76.77% of the respondents in our study indicated they participated in sport during the last two years of primary school, which is high but in accordance with Dutch national statistics [53]. Because we measured sport participation simply as yes or no, this means there is little variation in this measurement. Although the direction of the effect is as expected, the lack of variation could mean that we did not find a significant effect because most people participated in sport at that age. An alternative explanation could be that secondary school is a period in which young people stop playing sports (on average) [19,54,55]. Those who still participate in sport at the end of secondary school have already been able to maintain their sport participation in a phase in which resources are also changing (e.g., more time needed for homework, more influence from friends for other activities, etc.). They are more persistent in their sports participation.

Yet another explanation could also be that the majority of people who participated in sport at secondary school also did so during primary school, meaning a longer period of socialisation and habit building, resulting in continued sport participation after becoming a parent. Future research should first use a more detailed measure of sport participation (hours of sport, type of sport, etc.), which would result in more variation to determine if sport during primary school really has no effect on sport participation after becoming a parent. Then the alternative explanations can be examined in depth.

The second complication with socialisation has to do with the influence of parents in the last two years of secondary school. While seeing one’s parent(s) participate in sport during the last two years of primary school greatly decreases the likelihood of stopping participating in sport when people become parents, seeing one’s parent(s) sport during the last two years of secondary school has no effect at all. This could be explained by the fact that the older children become, the less important the influence of parents becomes and the more important the influence of friends becomes [56], especially during puberty, children want to break away from their parents [57]. Another explanation could be that seeing one’s parents participate in sport during primary school may be more of an example of what one wants to be like when one has children. At the end of secondary school, parents are older and in a different life phase compared to having young children. The role model effect may therefore be diminished. So future research should not only look at the sport behaviour of parents at the end of secondary school but also earlier. In addition, research should also examine the influence of the sport participation of peers to gain a better understanding of the influence of socialisation in puberty.

For the interpretation of the results, it is important to take the Western context of the current study into consideration. That is, in the Netherlands there is a sufficiently dense sport infrastructure, where availability of sport facilities is generally not a determinant of sport participation [58], and results cannot be generalized to other contexts. The design of the current study also leads to suggestions for future research. One limitation is that our data set only allowed us to examine continued vs. stopped sport participation. It would be informative to also include more subtle changes, such as sport frequency and sport context (club, group or individual), since those subtle changes also appear to be happening [13]. In addition, further research should not only focus on sport itself, as in this study, but on physical activity and sport. It could be the case that although new parents stop practising sport, they become more active in daily life by walking with their child. Or it could be even the case that new parents who did not participate in sport before they became a parent now comply with the exercise guidelines by walking with their child or doing other activities together. In addition, further research may examine to what extent physical activity and sport participation of new parents are related to other health-related lifestyle habits, such as nutrition, substance use such as tobacco or alcohol and rest/sleep and if these habits change in the same way as the sport participation. Finally, this study only focussed on the influence of resources and socialization. Factors such as personality [59], psychological aspects [60] and social networks [61] are neglected. Including all these factors in a single study would contribute to a better overall understanding of sport participation of new parents.

## 5. Conclusions

This study contributes to our understanding of participation in sport after becoming a parent for the first time. We replicated findings from several earlier studies by showing that resources (physical, temporal and social) and socialisation with sport affect sport participation. We extended previous findings by showing that resources and socialisation simultaneously affect sport participation. In line with this, further research is warranted to understand which factors help the socialisation aspect of practising sport. Since physical and temporal resources are difficult to change, socialisation could be the key to preventing new parents from stopping practising sport. Our study suggests that, to ensure future parents keep participating in sport, their parents should set a good example by sporting themselves.

## Figures and Tables

**Table 1 ijerph-21-00342-t001:** Descriptive statistics, *n* = 594.

Variable		% or M			SD	Range
		Overall	Not stopped	Stopped		
Stop practising sport	No	65.82%				
	Yes	34.18%				
Gender	Woman	68.01%	61.89%	79.80%		
	Man	31.99%	38.11%	20.20%		
Age at becoming a parent		28.87	29.12	28.40	3.35	18–38
Nationality	Dutch	90.07%	92.84%	84.73%		
	Non-Dutch	9.93%	7.16%	15.27%		
Educational level		20.90	21.04	20.64	1.99	12–23
Still recovering from giving birth	No	88.88%	92.33%	79.31%		
	Yes	12.12%	7.67%	20.69%		
Too tired	No	71.38%	78.52%	57.64%		
	Yes	28.62%	21.48%	42.36%		
Not enough time	No	47.47%	55.75%	31.53%		
	Yes	52.53%	44.25%	68.47%		
Supporting partner	No	57.24%	51.15%	68.97%		
	Yes	42.76%	48.85%	31.03%		
Babysitter	No	74.58%	72.89%	77.83%		
	Yes	25.42%	27.11%	22.17%		
Practising sport primary school	No	23.23%	19.95%	29.56%		
	Yes	76.77%	80.05%	70.44%		
Practising sport secondary school	No	35.02%	29.67%	45.32%		
	Yes	64.98%	70.33%	54.68%		
Parent(s) practising sport primary school	No	63.13%	57.78%	75.37%		
	Yes	36.87%	43.22%	24.63%		
Parent(s) practising sport secondary school	No	66.50%	61.64%	75.86%		
	Yes	33.50%	38.36%	24.14%		

**Table 2 ijerph-21-00342-t002:** Logit regression analyses on the likelihood of stopping practising sports after becoming a parent for the first time, *n* = 594.

		Model 1					Model 2					Model 3				
		OR	S.E.	*p*	95% CI		OR	S.E.	*p*	95% CI		OR	S.E.	*p*	95% CI	
*Personal characteristics*																
Gender	Woman = ref	0.501	0.113	**0.001**	0.322	0.780	0.489	0.111	**0.001**	0.313	0.764	0.532	0.123	**0.003**	0.338	0.838
Age at becoming a parent		0.959	0.029	0.079	0.904	1.016	0.954	0.029	0.060	0.900	1.012	0.956	0.029	0.072	0.900	1.015
Nationality	Dutch = ref	0.348	0.106	**0.001**	0.192	0.630	0.363	0.112	**0.001**	0.199	0.664	0.421	0.133	**0.003**	0.227	0.780
Educational level		0.867	0.043	**0.002**	0.787	0.956	0.869	0.043	**0.003**	0.788	0.959	0.884	0.046	**0.008**	0.799	0.978
*Physical resources*																
Still recovering from giving birth		2.941	0.844	**0.000**	1.676	5.163	2.861	0.822	**0.000**	1.628	5.026	3.117	0.921	**0.000**	1.746	5.563
Too tired		1.787	0.371	**0.003**	1.190	2.686	1.729	0.364	**0.005**	1.145	2.611	1.719	0.370	**0.006**	1.127	2.620
*Temporal resources*																
Not enough time		2.854	0.588	**0.000**	1.906	4.274	2.574	0.554	**0.000**	1.688	3.924	2.623	0.579	**0.000**	1.702	4.042
*Social resources*																
Supporting partner							0.632	0.137	**0.017**	0.413	0.966	0.628	0.139	**0.018**	0.407	0.970
Babbysitter							1.179	0.283	0.246	0.737	1.888	1.199	0.292	0.228	0.744	1.934
*Socialisation with sports*																
Practising sport primary school												0.945	0.231	0.408	0.585	1.524
Practising sport secondary school												0.571	0.124	**0.005**	0.373	0.875
Parent(s) practising sport primary school												0.436	0.143	**0.006**	0.229	0.830
Parent(s) practising sport secondary school												1.214	0.408	0.282	0.628	2.345
Constant		42.350	50.865	0.001	4.022	445.875	55.175	67.570	0.001	5.004	608.375	56.858	70.915	0.001	4.933	655.293
Pseudo R2		0.132					0.138					0.167				

## Data Availability

The data can be downloaded from the Dutch scientific data archive: 10.17026/dans-26k-ztw7.

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
