# Peer review of "Why Do New Parents Stop Practising Sport? A Retrospective Study towards the Determinants of Dropping Out after Becoming a Parent"

_ijerph, 2024, doi:10.3390/ijerph21030342_

Round 1

Reviewer 1 Report

Comments and Suggestions for Authors

The research in the manuscript addresses an important question: the reasons for dropping out of sport after becoming a parent. The aims and questions of the research are clear and theoretically well supported. It uses relevant sociological theories that are well suited to answering the research questions. In the fifth hypothesis, the authors assume the influence of a supportive partner, but it would be useful to examine the role of close family members (such as the support of parents/sisters, brothers, etc. of the parents). This helps to clarify the role of social support in the discussion. In relation to the hypotheses on the role of sport socialisation, there is a huge difference between participation in competitive and non-competitive sport. Children have different experiences in these forms of sport, which may influence their attitudes and willingness to participate in sport in later life. Therefore, the role of competitive and recreational sport participation should be investigated. 

Although the hypothesis is well supported theoretically, I am missing a summary chapter on the factors examined that may influence dropout from sport after becoming a parent.

The methodological background is described in detail. Why did the author decide to add the answer 'Yes, my partner enabled me to do some things, but not as much as I wanted' to 'No'?

CI should be added to the logistic regressions. The logistic regressions should be followed by two-variable analyses (crosstabulations). It would provide important information for readers to see the differences between sporting and non-sporting parents in percentages.

At the end of the discussion, it is important to present the limitations and strengths of the study.

Author Response

We like to thank the reviewer for his / her effort and his / her suggestions to make the manuscript better. Below we discuss the questions and suggestions of the reviewer. The questions / remarks of the reviewer are in black, our response is in red.

The research in the manuscript addresses an important question: the reasons for dropping out of sport after becoming a parent. The aims and questions of the research are clear and theoretically well supported. It uses relevant sociological theories that are well suited to answering the research questions. In the fifth hypothesis, the authors assume the influence of a supportive partner, but it would be useful to examine the role of close family members (such as the support of parents/sisters, brothers, etc. of the parents). This helps to clarify the role of social support in the discussion. In relation to the hypotheses on the role of sport socialisation, there is a huge difference between participation in competitive and non-competitive sport. Children have different experiences in these forms of sport, which may influence their attitudes and willingness to participate in sport in later life. Therefore, the role of competitive and recreational sport participation should be investigated. 

We agree with the reviewer that social support is not only provided by the partner, but can also be provided by other family members for example. In the questionnaire we asked about a babysitter, and did not further specify who this person is. As a result, this may include parents or siblings of the new parents, but we were not able to further scrutinize this in the result. We do include this aspect as possible explanation for non-significant effect of a babysitter and in the recommendations for future research.

With respect to participation in competitive or non-competitive sport, we did examine this for the sport that the new parents practice(d), but this did not lead to significant different. To keep the results as clear as possible, we did not include this in the final models. We did not have the information about the type of sport that was practiced when they were children. This may be an important aspect to delve deeper into the mechanism of socialization.

Although the hypothesis is well supported theoretically, I am missing a summary chapter on the factors examined that may influence dropout from sport after becoming a parent.

In accordance to the reviewer’s suggestion we added a summery on the factors examined that may influence dropout from sport after becoming a parent.

The methodological background is described in detail. Why did the author decide to add the answer 'Yes, my partner enabled me to do some things, but not as much as I wanted' to 'No'?

We want to thank the reviewer for his / her nice words about the method section. We decided to include support of the partner as a dichotomous variable (no – yes) for two reasons. First and foremost theoretically. Because people who answered ‘'Yes, my partner enabled me to do some things, but not as much as I wanted’ could still be limited in their sport activities. We want to be sure that we measure the correct partner effect and therefore we have a strict selection. Second, methodologically. Less than eight percent of our respondents indicated not having a partner at all, or indicated not having a supportive partner. This skewed distribution makes impossible to have a proper dichotomous division. When we included this variable with three categories (1) no partner or no support at all, 2) not as many support as wanted, 3) as many support as wanted) we found a non-significant influence of the first two categories and only a significant influence of the third category. For the parsimony of the models we decided to make a distinction between as many support as wanted versus the other categories.

CI should be added to the logistic regressions. The logistic regressions should be followed by two-variable analyses (crosstabulations). It would provide important information for readers to see the differences between sporting and non-sporting parents in percentages.

As suggested we added the CI to the tables. Also we extended Table 1 as suggested by the reviewer with crosstabulations for people who don’t stop and do stop participating in sports.

At the end of the discussion, it is important to present the limitations and strengths of the study.

As suggested by the reviewer, in the revised version we discuss the limitations of our study more explicitly.

Reviewer 2 Report

Comments and Suggestions for Authors

The study investigates an interesting aspect that significantly characterizes people's lifestyle and the perinatal health of new parents.

However, it should be underlined that the researchers take into consideration a Western context where research participants have the opportunity to use adequate sports facilities such as equipped parks, gyms and swimming pools.

The results highlighted in this study are certainly original and stand out as an important contribution to the literature.

In my opinion, however, the integration of some elements would make the work richer and more complete.

In the introductory part, as well as in the aspects investigated and in the Discussion section, there is a total lack of references regarding the psychological aspects that characterize the perinatal period and their possible impact on the choice to practice sports before and after the birth of a child. For example, what does the literature report on sporting activity in the conditions of planned/unplanned pregnancy, pregnancy at risk, mothers with premature birth, type of birth, breastfeeding, presence of twins. Furthermore, how much can the personality profile of new parents influence sports practice.

Finally, I suggest including a section reporting the limitations of the study in the Discussion section.

Author Response

We like to thank the reviewer for his / her effort and his / her suggestions to make the manuscript better. Below we discuss the questions and suggestions of the reviewer. The questions / remarks of the reviewer are in black, our response is in red.

The study investigates an interesting aspect that significantly characterizes people's lifestyle and the perinatal health of new parents.

However, it should be underlined that the researchers take into consideration a Western context where research participants have the opportunity to use adequate sports facilities such as equipped parks, gyms and swimming pools.

We agree with the reviewer that the context of the study is important for a good interpretation of the results. We now included this aspect in the final paragraph of the discussion.

The results highlighted in this study are certainly original and stand out as an important contribution to the literature.

In my opinion, however, the integration of some elements would make the work richer and more complete.

In the introductory part, as well as in the aspects investigated and in the Discussion section, there is a total lack of references regarding the psychological aspects that characterize the perinatal period and their possible impact on the choice to practice sports before and after the birth of a child. For example, what does the literature report on sporting activity in the conditions of planned/unplanned pregnancy, pregnancy at risk, mothers with premature birth, type of birth, breastfeeding, presence of twins.

We completely agree with the reviewer that these are important aspect which can affect sport behaviour of the parents, and specifically the mother. However, the scope of this article is to test two sociological theories against each other that could partly explain the reasons why people stop doing sports after they become a parent. Therefore, we only look at people who could stop doing sport (so who participated before becoming a parent). And that is the reason that we focus on aspects of socialization and resources. Although we look at the effect of gender, we did not include psychological aspects and the mentioned conditions of planned/unplanned pregnancy, pregnancy at risk, mothers with premature birth, type of birth, breastfeeding, presence of twins. We didn’t do so because we don’t have the information about these condition. In addition, the number of some conditions are so low in the Netherlands (premature birth in 2022 6.6% before 37 weeks of them and 0.15%  extreme premature; breastfeeding in the year a child is one year old, reliable figures are lacking in the Netherlands since registration if this stops when the child is 9 months, at this age less than 12% of the mothers do so, and twins < 1%) that we don’t mention it, also to keep the scope of the paper clear.

Furthermore, how much can the personality profile of new parents influence sports practice.

We totally agree with the reviewer that personality can have a big influence on stopping or continuing doing sports after becoming a parent. In our last model we only explain 16.7% (pseudo R2) of the variance. Meaning 83.3% of the variance is explained by other factors, such as personality. Although we believe that personality has a big influence, it doesn’t fit in the scope of our article; that is, test two sociological theories against each other that could partly explain the reasons why people stop doing sports after they become a parent. Studies towards personality and motivation and sporting behaviour (on different populations) showed that personality explains between 10 (Costa & Oliva, 2012) and 16 percent (Allen et al. 2013) of sport participation. To explain sport participation fully, we should take into account more factors. The focus of this article is to show the influence of socialization, resources and gender, however in the discussion we now discuss this limited scope and the need to look at more factors simultaneously.

Costa, S., & Oliva, P. (2012). Examining relationship between personality characteristics and exercise dependence. Review of psychology, 19(1), 5-11.

Allen, M. S., Greenlees, I., & Jones, M. (2013). Personality in sport: A comprehensive review. International Review of Sport and Exercise Psychology, 6(1), 184-208.

Finally, I suggest including a section reporting the limitations of the study in the Discussion section.

Although we mentioned limitations in the previous discussion, we now mention them more clearly in the revised version, as also mentioned as response to the final comment of reviewer 1.

Reviewer 3 Report

Comments and Suggestions for Authors

Dear authors. First of all, I would like to congratulate you on the article presented. I believe that understanding the phenomenon of sport dropout is vital to avoid behaviours that lead to physical inactivity and sedentary lifestyles. Therefore, I am sending you some suggestions that I hope you will accept as a means of improving the article presented.

Abstract: I miss the main objective formulated in such a way that it can be evaluated with the proposed methodology and that somehow leads us to the conclusions obtained. In the same way, a conclusion and main results are usually highly recommended when writing the abstract of a publication.

Introduction: 

Line 28: I believe that conceptually the practice of Physical Activity (as a hierarchically superior action to sport) and being active are the provokers of health benefits (as a global concept). I would therefore change the term sport to Physical Activity and Sport. In the same way I would add bibliography to reinforce the initial and important concept in the presented paper. As in line 38

Line 40: I think that using the leap from Primary to Secondary as an argument towards vital changes and then talking about marriage would not be progressive or equivalent, perhaps the leap to University could be sequentially closer, therefore reviewing this article on abandonment of Physical Activity by university students could be chronologically closer (https://doi.org/10.3390/ijerph18115721).

Line 63: at this point it would be essential to formulate the objective of the study. From the objective we would formulate the research question(s).

Line 88: when formulating the first hypothesis and talking about physical resources, especially in the case of women, I would make two changes. On the one hand, I would talk about physical condition rather than physical resources, I can understand that time or social networks are considered resources, but I do not conceive of physical condition. On the other hand, more and more countries are implementing pre-natal and postpartum physical activity programmes as it has been proven that physical exercise is protective against the physiological events that cause postpartum: gestational weight retention, episodes of urine leakage, postpartum depression, etc.

Line 110: regarding the time resource, it would be interesting to compare subjects who are previously active or highly active first-time parents with others who are less active. In the same way, having a baby can lead to long walks and hikes that help to meet the recommendations for being considered active by taking 10,000 or more steps per day (https://pubmed.ncbi.nlm.nih.gov/14715035/).

The whole theoretical framework is conditioned by the adherence to healthy lifestyle habits of the participants. These include: physical activity, nutrition, substance use such as tobacco or alcohol, and rest/sleep. 

I would like to be able to compare first-time parents with good habits with others who do not adhere to them.

Materials and method:

Line 193 Once Table 1 is named I would present it there to first look at the data and then be able to read the results. 

Results:

I do not feel qualified to evaluate the analyses performed as I am not proficient in linear regressions, but would it be interesting to present a correlation matrix?

Discussion:

It is well developed and all the variables presented are analysed.

Author Response

We like to thank the reviewer for his / her effort and his / her suggestions to make the manuscript better. Below we discuss the questions and suggestions of the reviewer. The questions / remarks of the reviewer are in black, our response is in red.

Dear authors. First of all, I would like to congratulate you on the article presented. I believe that understanding the phenomenon of sport dropout is vital to avoid behaviours that lead to physical inactivity and sedentary lifestyles. Therefore, I am sending you some suggestions that I hope you will accept as a means of improving the article presented.

Abstract: I miss the main objective formulated in such a way that it can be evaluated with the proposed methodology and that somehow leads us to the conclusions obtained. In the same way, a conclusion and main results are usually highly recommended when writing the abstract of a publication.

We agree with the reviewer that the main result was not presented clearly in the abstract. We therefore added more information about the results. In addition, we adjusted the wording in the abstract in order to be more clear about the match between objective, method, results and conclusion.

Introduction: 

Line 28: I believe that conceptually the practice of Physical Activity (as a hierarchically superior action to sport) and being active are the provokers of health benefits (as a global concept). I would therefore change the term sport to Physical Activity and Sport. In the same way I would add bibliography to reinforce the initial and important concept in the presented paper. As in line 38

As suggested we changed the terminology in line 28 and we added some references in line 38.

Line 40: I think that using the leap from Primary to Secondary as an argument towards vital changes and then talking about marriage would not be progressive or equivalent, perhaps the leap to University could be sequentially closer, therefore reviewing this article on abandonment of Physical Activity by university students could be chronologically closer (https://doi.org/10.3390/ijerph18115721).

We would like to thank the reviewer for this wonderful addition. We included this in our manuscript.

Line 63: at this point it would be essential to formulate the objective of the study. From the objective we would formulate the research question(s).         

At line 60 we make clear what the objective of our study is.

Line 88: when formulating the first hypothesis and talking about physical resources, especially in the case of women, I would make two changes. On the one hand, I would talk about physical condition rather than physical resources, I can understand that time or social networks are considered resources, but I do not conceive of physical condition. On the other hand, more and more countries are implementing pre-natal and postpartum physical activity programmes as it has been proven that physical exercise is protective against the physiological events that cause postpartum: gestational weight retention, episodes of urine leakage, postpartum depression, etc.

We followed the suggestion of the reviewer and make clear that physical resources, especially in the case of women who gave birth, are related to their physical condition. Mention the subheading, ‘Physical resources: Being physically able to practise sport after becoming a parent’ in which we make clear that it is about physically being able to practise sport. However, since we also focus on sleep deprivation and energy we follow the work of Shilling (2004) which sees the human body as physical capital and thus a resource.

Because of the big physical impact, especially for women, for getting a child we focused on sport participation in the year that their first child was one year old (and we excluded women we were pregnant again). Be doing this and including the condition of ‘not being fully recovered’ for women we tried as good as possible to recon for the physical challenges women has.

Shilling*, C. (2004). Physical capital and situated action: A new direction for corporeal sociology. British journal of sociology of education, 25(4), 473-487.

Line 110: regarding the time resource, it would be interesting to compare subjects who are previously active or highly active first-time parents with others who are less active. In the same way, having a baby can lead to long walks and hikes that help to meet the recommendations for being considered active by taking 10,000 or more steps per day (https://pubmed.ncbi.nlm.nih.gov/14715035/).

We agree with the reviewer that this is a very interesting question. However with the current data we are not able to answer this question. We didn’t ask retrospectively about active behaviour, such as walking, since this is less tangible and not as easy to remember as sport behaviour. As a result, these answers would not be reliable enough. However, the current article is based on the first wave of a three wave data collection. The goal of the data collection is that we include people who make the transition into parenthood during these waves. For these participants, we will have detailed information about their physical activity and sport before and after they become a parent. This makes it possible to answer the questions if people who drop out from sport after they become a parent become more active outside of sport. And that people who don’t sport before they become a parent get more active outside of sport. However, this is still work in process, since we recently finished the data collection.          

The whole theoretical framework is conditioned by the adherence to healthy lifestyle habits of the participants. These include: physical activity, nutrition, substance use such as tobacco or alcohol, and rest/sleep. 

I would like to be able to compare first-time parents with good habits with others who do not adhere to them.

We completely agree with the reviewer that it would be wonderful to compare first time parents with good / healthy habits with first time parents who haven’t these habits. Unfortunately, our data doesn’t allow us to do so. We only have information about parents sport behaviour. However, in the discussion we do suggest this for further research.

Materials and method:

Line 193 Once Table 1 is named I would present it there to first look at the data and then be able to read the results. 

We changed this according to the suggestion of the reviewer.

Results:

I do not feel qualified to evaluate the analyses performed as I am not proficient in linear regressions, but would it be interesting to present a correlation matrix?

There can be differences between what correlations show and the results of a regression analysis, since regression analyses shows the relationship between the variables under control for each other. Therefore, we don’t want to show the correlation matrix. However, as suggested by reviewer 1 in Table 1, we present the descriptive statistics overall and for people who did and did not stop doing sport. This is, in our opinion, more informative in combination with the regression analyses than showing the correlation matrix.

Discussion:

It is well developed and all the variables presented are analysed.

We would like to thank the reviewer for this compliment.